# Visualization of Bacterial Protein Complexes Labeled with Fluorescent Proteins and Nanobody Binders for STED Microscopy

**DOI:** 10.3390/ijms20143376

**Published:** 2019-07-10

**Authors:** Kimberly Cramer, Anna-Lena Bolender, Iris Stockmar, Ralf Jungmann, Robert Kasper, Jae Yen Shin

**Affiliations:** 1Max Plank Institute of Biochemistry, 82152 Martinsried, 82152 Munich, Germany; 2Faculty of Physics and Center for Nanoscience, Ludwig Maximilian University, 80539 Munich, Germany; 3Max Plank Institute of Neurobiology, 82152 Martinsried, 82152 Munich, Germany

**Keywords:** STED, bacteria, super-resolution microscopy, fluorescent proteins, nanobody, cell division

## Abstract

In situ visualization of molecular assemblies near their macromolecular scale is a powerful tool to investigate fundamental cellular processes. Super-resolution light microscopies (SRM) overcome the diffraction limit and allow researchers to investigate molecular arrangements at the nanoscale. However, in bacterial cells, visualization of these assemblies can be challenging because of their small size and the presence of the cell wall. Thus, although conceptually promising, successful application of SRM techniques requires careful optimization in labeling biochemistry, fluorescent dye choice, bacterial biology and microscopy to gain biological insights. Here, we apply Stimulated Emission Depletion (STED) microscopy to visualize cell division proteins in bacterial cells, specifically *E. coli* and *B. subtilis*. We applied nanobodies that specifically recognize fluorescent proteins, such as GFP, mCherry2 and PAmCherry, fused to targets for STED imaging and evaluated the effect of various organic fluorescent dyes on the performance of STED in bacterial cells. We expect this research to guide scientists for in situ macromolecular visualization using STED in bacterial systems.

## 1. Introduction

The expression of targets of interest fused to fluorescent proteins (FPs) is one of the labeling approaches utilized to indirectly or directly visualize proteins with diffraction-limited and super-resolution microscopy (SRM) such as stimulated emission depletion (STED) [1,2], structured illumination microscopy (SIM) [3,4], DNA points accumulation for imaging in nanoscale topography (DNA-PAINT) [5], (direct)stochastic optical reconstruction ((d)STORM) [6,7] and photoactivatable localization (PALM) [8] microscopies. While imaging of targets in bacterial cells has mostly used genetically encoded fluorescent proteins for direct visualization, indirect visualization of FPs using binders with organic dyes could provide higher versatility and higher spatial resolution due to often superior photophysical properties [9]. However, in the latter approach, it remains challenging to achieve a high labeling efficiency of intracellular proteins because of the limited cell wall permeability [9,10] (see Figure 1).

The direct visualization of FPs, expressed as fusion proteins in bacterial targets, has been widely implemented in super-resolution light microscopies [9,11,12]. One of the most beneficial aspects of this method is that bacterial samples can be directly imaged—even live—without intensive sample preparation. Using direct visualization of FPs with SRM, researchers have observed protein assemblies, such as the cell division machinery [9,13,14,15,16,17], membrane microdomains [18,19,20], and the cytoskeleton [21,22] in various bacterial organisms such as *Escherichia coli, Bacillus subtilis, Staphylococcus aureus* and *Caulobacter crescentus*. In addition, recent developments of dual-color imaging using FPs for SIM [15,23] and STED [24] have led scientists to gain biological insights into the relationship between the ultrastructure of protein assemblies and their function, which would otherwise not been accessible.

On the other hand, imaging immunolabeled samples using super-resolution microscopy has been performed to a lesser extent in bacterial cells, most likely due to the limited labeling efficiency of intracellular proteins given by low cell wall permeability [9,10]. In this indirect visualization method, binders, i.e., primary antibodies that bind the target of interest, followed by secondary antibodies carrying a fluorescent molecule, need to enter bacteria. For binders to successfully reach intracellular targets, the cell wall must be at least partly digested using enzymes such as lysozyme. Although limited, there are a few examples in the literature that implemented antibodies to visualize bacterial proteins with SRM. For instance, FtsZ, one of the most essential cell division proteins, was visualized with STED and SIM microscopy using primary and secondary antibody binders in *B. subtilis* cells [25,26]. Most recently, two different cell division proteins, FtsZ and FtsN, were simultaneously visualized using antibody binders in *E. coli* cells [24].

Although valuable, indirect immunolabeling using primary and secondary antibodies increases the apparent size of the visualized structure or introduces a localization bias of 10–20 nm when using SRM [27,28,29]. One way of reducing the distance between the target of interest and fluorescent label (linkage error) is by using significantly smaller binders, such as nanobodies (~2 nm) or FAB fragments (a smaller version of an antibody) [30]. To this extent, Vedyaykin et al. visualized FtsZ in *E. coli* cells using a conventional primary antibody and a secondary FAB fragment with STORM [31]. Recently, the use of dye-labeled nanobodies as nanoscale detection tools has been implemented to visualize protein complexes in eukaryotic cells with SRM [30].

Another crucial aspect to consider when performing SRM is dye properties. Some properties (e.g., high brightness, photostability, low phototoxicity) are universally desired among all SRM techniques, however, some specific properties are of higher or lower importance depending on the imaging modality. For instance, STORM requires dyes that blink, i.e., switch between fluorescence ON- and OFF-states, such as Cy5 derivatives [32]. For STED, however, it is advantageous that dyes do not blink [33]. Additionally, a particular property of a dye (e.g., hydrophobicity, net charge) might influence the specificity of a binder. Thus, the identification of a functional combination of dyes and binders most likely depends on the organism under investigation and even upon the target of interest.

Despite great strides in bacterial SRM, research using the direct visualization of targets greatly outweighs that using indirect visualization methods, which is mainly due to the comparably more complex sample preparation requirements and the limited availability of good binders. In this study, we sought to assay labeling approaches for STED microscopy, increase the number of imageable targets in *B. subtilis* by using nanobody binders that recognize fluorescent proteins, such as green fluorescent protein (GFP) or red fluorescent protein (RFP) and their derivatives, and identify combinations of binders and dyes that are suitable for STED imaging in bacterial cells.

## 2. Results

### 2.1. Nanobodies Recognizing Fluorescent Proteins Enable Visualization of Target Proteins in Bacteria with STED Microscopy

Targets under investigation can be visualized indirectly using binders that specifically detect targets or fluorescent proteins fused to a target. Here, we implement a visualization method based on nanobodies that bind FPs for conventional confocal and STED microscopy (Figure 1). Our workflow comprises three main steps: (1) evaluation of whether FPs fused to target proteins are innocuous to the target protein function; (2) optimization for cell-wall permeabilization; (3) visualization of the target proteins using fluorescently labeled nanobodies.

To establish this, we chose DivIVA as a candidate protein. DivIVA is a cell division protein in the Gram-positive model bacterium *B. subtilis*, and its ultrastructure can only be visualized with super-resolution microscopy [14,34]. *B. subtilis* expressing either GFP or mCherry2 [35] fused to DivIVA and showed a fluorescent band at the division septa when imaged using diffraction-limited microscopy (Figure 2a,b). As expected, these proteins showed double bands (hereafter referred to as “DivIVA dual band”) when visualized with SIM microscopy. The distances between the two bands (ranging from ~80 nm to ~200 nm) were similar to previous reports (Figure 2a,b, Appendix A) [14,34].

Unlike eukaryotic cells, bacterial cells contain a cell wall that impedes the intracellular delivery of exogenous molecules, in our case nanobodies conjugated to fluorophores, potentially “trapping” these molecules and preventing their intracellular delivery. Thus, to efficiently deliver molecules, we first optimized the cell wall digestion step by treating fixed cells with various concentrations of lysozyme and delivering a fluorescently labeled binder that recognizes FPs. Specifically, we employed *B. subtilis* strains expressing photoactivatable mCherry (PAmCherry) fused to the DivIVA protein. The condition in which the cells presented the highest fluorescent signal from the nanobody at the cell division septa was considered the best for cell-wall permeabilization (Appendix A). The optimal cell permeabilization condition might differ from species to species and even strain to strain.

Visualization of GFP and mCherry2—fused to DivIVA—using the respective Atto647n-conjugated nanobody binders (NB^GFP^-Atto647n and NB^RFP^-Atto647n) shows a similar DivIVA dual-band localization pattern when compared to the direct visualization of FPs (Figure 2). These results indicate that both binders specifically bind the corresponding fluorescent proteins. As expected from our confocal imaging results and the property of Atto647n for STED imaging [36], Atto647n was suitable to image bacterial proteins and resolve the dual band of DivIVA with STED microscopy. In contrast, confocal microscopy was not able to resolve the dual band of DivIVA (Figure 2c,d). As a control experiment, we compared STED performance for organic dye (NB^RFP^-Atto647n) and fluorescent protein, specifically mCherry2. Our results showed that the use of organic dye, specifically Atto647n (i.e., NB-Atto647n) significantly enhanced the STED signal (Appendix A).

Thus, our established protocol using binders for RFP and GFP allowed us to efficiently image bacterial protein complexes with STED microscopy (Figure 2 and Figure 3). Although we could clearly resolve the DivIVA dual band at the division septa using both binders, the cellular background appeared higher upon visual inspection when using the RFP binder, NB^RFP^-Atto647n (Figure 3). To quantify this background, we determined the fluorescent signal at the cell division septa (A) and outside of the septa (B) (Appendix A), and used these values to calculate a signal-to-background ratio (A/B) and the percentage of the cellular background (B/A*100). Interestingly, NB^RFP^-Atto647n exhibited three times more background than the GFP binder, NB^GFP^-Atto647n (Table 1).

### 2.2. Evaluation of Unspecific Binding for GFP and RFP Nanobodies in Bacterial Cells

We reasoned that the higher cellular background of the RFP binder could be due to either (1) higher cytoplasmic contents in actual DivIVA-mCherry2 molecules compared to the *B. subtilis* strain expressing the DivIVA-GFP protein, or (2) unspecific binding of the NB^RFP^-Atto647n. To rule out these two possibilities, we quantified, as described above, the background fluorescence of mCherry2 and GFP in bacillus strains that either expressed DivIVA-mCherry2 or DivIVA-GFP. Contrary to the cellular background observed in STED images, both strains showed similar backgrounds of approximately 40% and 35% for DivIVA-GFP and DivIVA-mCherry2, respectively (Appendix A). Interestingly, the RFP binder, NB^RFP^-Star600, also recognized the PAmCherry fusion protein, although with a higher cellular background than the mCherry2 fusion protein (Table 1). Altogether, these results indicate that (1) both the GFP and RFP binders are suitable for STED microscopy, (2) the GFP binder presents higher specificity to GFP than the RFP binder does to mCherry2, and (3) the RFP binder binds unspecifically to *B. subtilis* cells.

### 2.3. Evaluation of Fluorescent Dyes for STED Imaging in Bacterial Cells

Next, we systematically evaluated the suitability and performance of various dyes to image proteins in bacteria using STED microscopy. Furthermore, we investigated the effect of the dye on unspecific nanobody binding. To this end, we imaged DivIVA-mCherry2 or DivIVA-GFP expressing bacillus strains using nanobodies conjugated with various fluorophores that are reported to be suitable for STED microscopy. However, these dyes have mostly been evaluated for suitability in eukaryotic cells [36,37]. Specifically, we employed Atto647n, Atto594, Star600 and Star635p dyes (properties compiled in Appendix A) conjugated either to the RFP or GFP nanobodies.

Our STED results show that NB^GFP^-Atto647n produced images with at least four times higher signal-to-background than the RFP binders conjugated with the same dye (Table 1). Interestingly, the background for the GFP nanobody altered when conjugated with different dyes. The cellular background when using NB^GFP^ increased as follows: Atto647n < Star635p < Atto594 (Figure 4, Table 1). This result indicates that all the tested dyes induced unspecific binding of NB^GFP^ in *B. subtilis* cells. In contrast, we did not observe significant differences in the cellular background for the NB^RFP^ when conjugated with different dyes (Figure 4, Table 1), indicating that the specificity of the NB^RFP^ is less influenced by the dyes compared to the GFP nanobody. Note that the spectral overlap between Atto594 and mCherry2 might have increased the signal-to-background ratio (Table 1). However, STED microscopy directly observing the mCherry2 fusion protein qualitatively showed a poorer signal-to-background ratio (Appendix A).

## 3. Discussion

In this study, we implemented for the first time a methodology for STED super-resolution microscopy to visualize bacterial protein complexes using nanobodies that bind fluorescent proteins. Interestingly, good STED dye performers for eukaryotic cells were not necessarily equally good for bacterial cells. Although we implemented nanobody-based STED in bacterial cells, we expect this research to be useful to the implementation of experimental design and sample preparation workflow for other species containing cell walls such as yeasts, plant cells and archaea.

To assay our approach, we visualized FtsZ and DivIVA cell division proteins because both form assemblies that are well described with super-resolution microscopies, e.g., SIM, STED, PALM and STORM [9,12,14,25]. Our super-resolved DivIVA dual band presented similar dimensions (Appendix A) to previously reported values [14,34]. However, results indicate different degrees of unspecific binding (background values in Appendix A), most likely due to the properties of the organic dye attached to the nanobodies. All nanobodies employed here resolved the DivIVA dual band (Figure 2, Figure 3 and Figure 4). In good agreement with the literature, our STED protocol also visualized the FtsZ protein localized in a “patchy” distribution along the circumference of the cell division plane in *E. coli* cells (Appendix A) [12,25].

Although STED can be performed in live bacteria [20,24,38,39], our aim was to establish a method based on immunolabeling, since organic fluorescent dyes exhibit higher photostability compared to fluorescent proteins [38]. Consequently, these dyes tolerate higher STED beam intensities, which is directly related to the resolution that STED microscopy can achieve [37,39]. The benefits of organic dyes in comparison to FPs were further demonstrated here when STED images of mCherry2 and NB^RFP^-Atto647n were directly compared in the same cell (Appendix A). In addition, development efforts have increased the number of suitable STED dyes for cell imaging [37,38], which, when used in combination with the immunolabeling method, offers higher versatility. In this regard, the ideal dye–nanobody pair should be innocuous to the specificity of the nanobody. However, this appears to not always be the case, as we have shown here (Figure 4 and Table 1). Additionally, the performance and suitability of a dye–nanobody pair might vary according to the targets and organisms under investigation. For example, while Atto594 dye performs well for eukaryotic cell imaging [40,41], it performed poorly in bacterial cell imaging (Figure 4 and Table 1). However, Atto647n performed equally well in bacterial cells, as previously reported for imaging of eukaryotic targets [37].

Our immunolabeling approach should be particularly interesting to labs that already have large strain collections of organisms expressing target proteins fused to FPs. Additionally, the growing repertoire of nanobodies can be used not only for STED microscopy, but also other imaging modalities such as STORM [30]. Furthermore, one could also employ nanobodies that directly or indirectly recognize their targets using primary and secondary nanobodies. Likewise, direct and indirect methods using FPs, nanobodies, and combinations of primary and secondary antibodies can be combined to make imaging much more versatile and used for implementation of multi-color imaging.

Finally, and most importantly, the presence of the cell wall in bacteria is an essential organelle to consider since it must be digested for the intracellular delivery of binders. As shown in this study, identifying a digestion condition that favors the delivery of exogeneous molecules while preserving cell morphology allowed us to visualize protein assemblies, namely DivIVA and FtsZ, with STED microscopy (Figure 2, Appendix A). Additionally, another factor to consider is identifying suitable and better performing fluorescent dyes for a particular target and the organism under investigation. The literature on STED microscopy for bacterial cells is much more limited compared to the literature available on STED microscopy for eukaryotic cells. Ideally, dyes should be innocuous to the specific binding of nanobodies and antibodies to their targets. However, this is not always the case, as it was shown that coupling Star635p or Atto594 dye to the GFP nanobody increases unspecific binding (compare cellular backgrounds in Table 1). Thus, we expect our table of signal-to-background ratio and the cellular background to be useful in improving experimental design (Table 1, Appendix A).

## 4. Materials and Methods

### 4.1. Reagents and Cell Culture

Bacterial strains used in this study are listed in the Appendix A. Luria Bertani broth, and SMG ([15 mM (NH4)2SO4, 61 mM K2HPO4, 44 mM KH2PO4, 3.4 mM sodium citrate 2xH2O, 1.7 mM MgSO4, 5.9 mM glutamate and 27 mM glucose] supplemented with 1.0 mM tryptophan) were used to grow bacteria. The cells were fixed with paraformaldehyde (P6148, Merck, Kenilworth, NJ, USA), immobilized with poly-L-lysine (Sigma P8920, St. Louis, MO, USA), and permeabilized with lysozyme (ThermoFisher, 90082, Waltham, MA, USA). PBSG (PBS + 20nM glucose) and PBST (PBS + 0.02% Tween20) were used for washing. ProLong Diamond Antifade Mountant (ThermoFisher, P36965, Waltham, MA, USA) or Abberior Mount Liquid Antifade (MM-2009, Abberior Instruments, Göttingen, Germany) were used as mounting media.

### 4.2. SIM Imaging

#### 4.2.1. Bacterial Sample Preparation

*Cell Growth.* Strains BHF3 and 1803 were streaked onto LB plates. Single colonies were grown overnight in LB medium at 30 °C, 220 rpm.

*Live Cell Sample Preparation.* Strain 1803 was inoculated 1:100 into fresh LB medium the next morning and grown at 30 °C and 150 rpm until OD_600_ = 0.5. An amount of 200 μL of cell culture was centrifuged for 20–30 s at 2000 xg. A cellular pellet was resuspended in 3 μL of LB medium and spotted on a 1.5% (*w*/*v*) agarose pad. A glass coverslip was placed on the agarose pad and cells were immediately imaged.

*Fixed Cell Sample Preparation*. Strain BHF3 was grown, fixed, and immobilized as performed in Stockmar et al. (2018). An amount of 2% paraformaldehyde was used for fixation. The maximum cellular density per fixation reaction was OD_600_ = 0.25. Cells were immobilized in multi-well chambers (μ-Slide Well Glass Bottom, Ibidi 80827, Gräfeling, Germany).

#### 4.2.2. SIM Data Acquisition and Processing

SIM images were acquired with a commercial Zeiss Elyra PS.1 microscope (Zeiss, Oberkochen, Germany) using an PCO pco.edge 4.2 m sCMOS Camera. An alpha Plan-Apochromat 100x/1,46 Oil DIC objective lens was used for fixed cell imaging and a C-Apochromat 63x/1,2 W Corr objective lens for live cell imaging. Images of strains BHF3 and strain 1803 appearing in Figure 2 are individual slices within a 2D z-stack. Exposure time was 200 ms for both GFP and mCherry2 imaging. GFP and mCherry2 were excited with a 488 nm OPSL Diode laser and a 561 nm OPSL Diode laser, respectively. Image analysis was done using Zeiss ZEN 2.1 software (Zeiss, Oberkochen, Germany). A noise filter of −3.2447 with a Max Isotrop was applied on the DivIVA-GFP image (Figure 2a), and a noise filter of −5 with a Max Isotrop of 1 was applied for the DivIVA-mCherry2 image (Figure 2b).

### 4.3. STED Imaging

#### 4.3.1. Bacterial Sample Preparation

*Cell Growth.* Strains 1803 and BHF3 were grown as previously described [34], except that the cells were grown overnight in LB medium.

*Fixation and Immobilization.* The cells were fixed as described in Section 4.2, via SIM Imaging. The maximum cellular density per fixation reaction was OD_600_ = 0.25. The fixed cells were immobilized on coverslips that had been incubated for 30 min with 0.01% poly-L-lysine solution and washed three times with Milli-Q water. Amounts of 150–200 μL of fixed cells containing a cellular density of ~OD_600_ 0.8 were added to each coverslip and left to settle for 30 min. The cells were then gently washed three times with PBSG.

*Immunolabeling Protocol.* The immobilized cells were permeabilized using 0.2 mg/mL lysozyme in PBSG for 5 min at 30 ºC, then immediately blocked for 1 h in 2% BSA. Nanobody binders were incubated overnight at 4 ºC. The following day, the cells were washed three times with PBST. All binders with their corresponding dilutions, targets and figures are found in Appendix A. Abberior Mount Liquid Antifade or ProLong^®^ Diamond Antifade Mountant was then added to glass microscope slides and coverslips were placed on top. After 30 min, the slides were sealed with clear or tan nail polish.

#### 4.3.2. STED and Confocal Data Acquisition and Processing

Confocal and STED images were acquired using a STEDYCON nanoscope system (Abberior Instruments, Göttingen, Germany) mounted on a Leica DMR X2 microscope and equipped with a specialized STED 100x oil immersion objective, NA 1.4 (Leica Microsystems). The excitation wavelengths were used according to the dye of the fluorescent protein specification: GFP with 488 nm, Atto594 and STAR600 with 561 nm, and 640 nm for Atto647N and Star635P. For both emission channels, a depletion laser at 775 nm was used. Fluorescent signals were detected on 3 separate APD detectors using standard bandpass filters (APD1: 650–700 nm, APD2: 575–625 nm, APD3: 505–545 nm) and a gated detection window starting at 1 ns after the laser pulse and closing after 6 ns. Finally, 2D STED images as single planes or as z-stacks with a slice distance of 200 nm were recorded and regions of interest were identified and processed using Fiji software [42,43] and OriginPro 9.1G (OriginLab, Northampton, MA, USA).

## Figures and Tables

**Figure 1 ijms-20-03376-f001:**
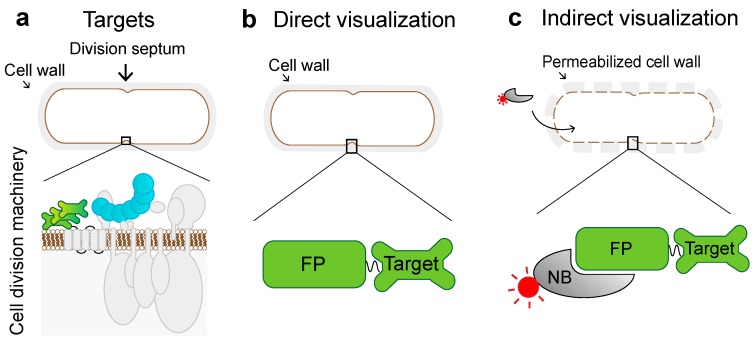
Schematic diagram of the bacterial protein complex and methods for visualization. (**a**) Cartoon representation of the *B. subtilis* cell featuring cell wall (gray), cell membrane (brown), and cell division proteins forming the cell division machinery. Proteins FtsZ and DivIVA visualized in this study are highlighted in blue and green, respectively. (**b**) The signal from a fluorescent protein (FP) fused to the target protein is directly visualized or (**c**) the target is indirectly visualized by fluorescently (red) labeled nanobodies (NB).

**Figure 2 ijms-20-03376-f002:**
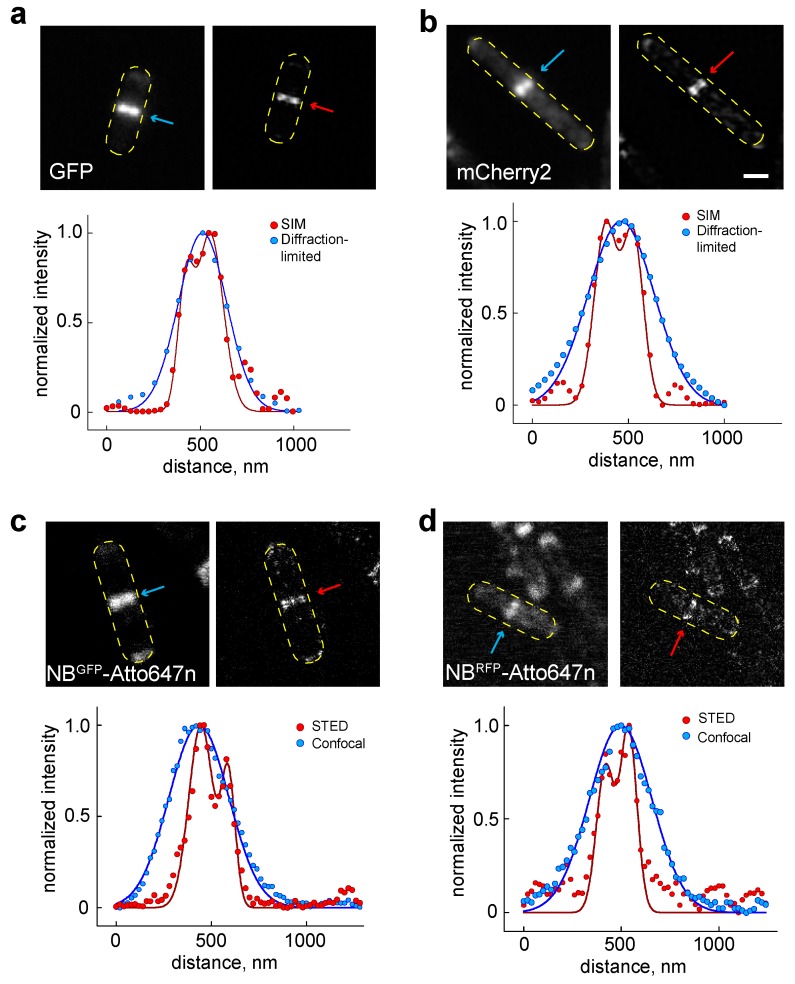
Visualization of the cell division protein DivIVA using SIM and STED microscopy. (**a**,**b**) Fluorescent signal from cells expressing either GFP or mCherry2 fused to DivIVA was imaged using diffraction-limited (left panel) and SIM (right panel) microscopies. Intensity profile of the signal along the longitudinal axis of the cell is shown in the lower panel. (**c**,**d**) Fluorescent signal from cells incubated with the Atto647n conjugated to either GFP nanobody (NB^GFP^-Atto647n) or RFP nanobody (NB^RFP^-Atto647n) was imaged using a conventional confocal (left panel) and STED (right panel) microscopy. The intensity profile of the signal along the longitudinal axis of the cell is shown in the lower panel. Scale bar 1 μm.

**Figure 3 ijms-20-03376-f003:**
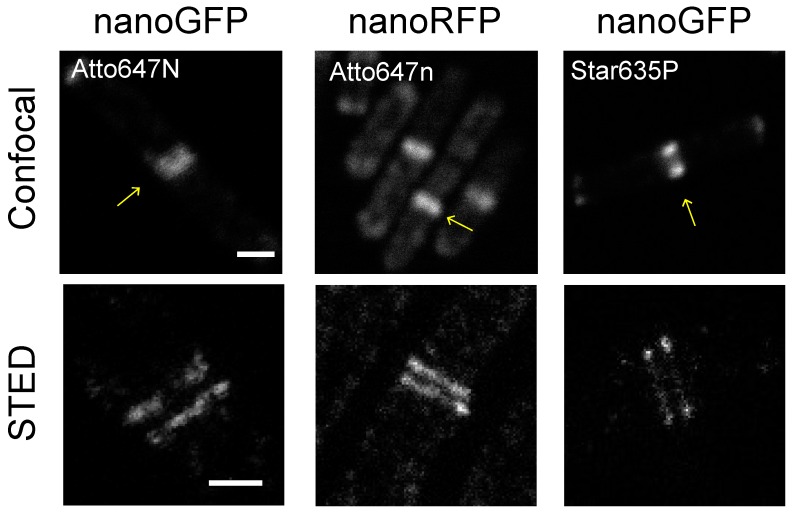
Comparison of Atto647n and Star635p for bacterial STED imaging. *B. subtilis* cells expressing either DivIVA-GFP or DivIVA-mCherry2 were visualized using nanobodies conjugated with Atto647n or Star635p. STED images show an enlarged field of view of the object marked with an arrow in the confocal image. Scale bars 1 μm and 0.5 μm, for confocal and STED images, respectively.

**Figure 4 ijms-20-03376-f004:**
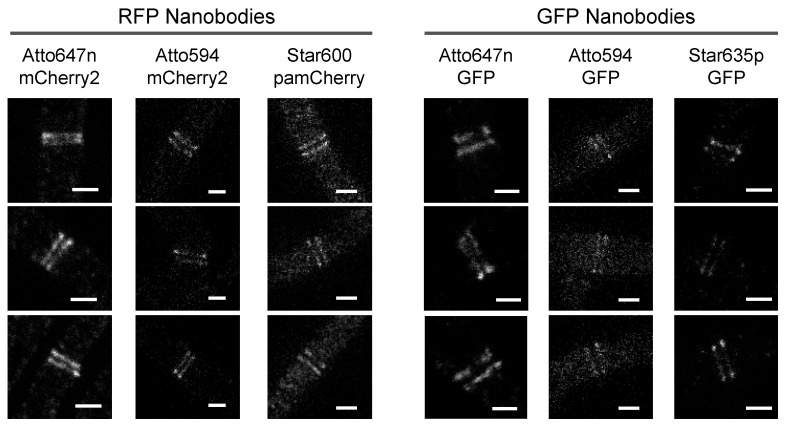
STED images of DivIVA dual bands visualized with nanobodies. *B. subtilis* cells expressing GFP, mCherry2 or PAmCherry fused to DivIVA were imaged using nanobodies containing the indicated organic fluorescent dyes. Scale bars 500 nm.

**Table 1 ijms-20-03376-t001:** Summary of Dyes Performance for STED Imaging in Bacterial Cells.

Nanobody-Dye	Target	Signal-to-Background Ratio	Cellular Background
NB^GFP^-Atto647n	GFP	12	11 %
NB^RFP^-Atto647n	mCherry2	3	36 %
NB^GFP^-Atto594	GFP	n.a. *	n.a. *
NB^RFP^-Atto594	mCherry2	3	38 %
NB^RFP^-Star600	PAmCherry	2.4	42 %
NB^GFP^-Star635p	GFP	3	34 %

All targets were fused to DivIVA protein. Signal-to-cellular background ratio and cellular background were calculated as described in the main text and in Appendix A. * not applicable (n.a.), values corresponding to signal and cellular background could not be determined due to high cellular background.

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
