# Peer review of "Visualization of Bacterial Protein Complexes Labeled with Fluorescent Proteins and Nanobody Binders for STED Microscopy"

_ijms, 2019, doi:10.3390/ijms20143376_

Round 1

Reviewer 1 Report

The authors developed a protocol for the STED imaging of bacterial proteins. Instead of using the emission of fluorescent proteins (FPs) for imaging, they used the signal of fluorescently labeled nanobodies binding specifically to FPs. The labels applied have higher photostability than FPs, affording higher beam intensities which can lead to higher resolution. The main difficulty of the protocol is presumably the partial digestion of the cell wall. The applicability of the protocol was demonstrated on STED images of cell division proteins in bacterial cells. The quality of the images obtained with different target FP – (nanobody)-dye combinations have been compared.

The results presented in this manuscript may be helpful in the development of super-resolution microscopic methods for the imaging of bacteria and other species with cell walls. The paper is written thoroughly. I recommend it for publication.       

Author Response

We thank Reviewer #1 for the positive comments on the submitted manuscript, and are pleased to see that the main messages of the manuscript were clear. Additionally, we are happy the Reviewer shares our view that the results could be helpful for further super-resolution microscopy development in cell wall containing species.

Reviewer 2 Report

- This publication proposes a method based on nanobodies, tagged with efficient dyes for STED imaging, that bind specifically to fluorescent proteins to image bacterial proteins. 3 steps are presented, including: preparation of FP fused target proteins (and verification of their biological activity after modification), cell permeation, imaging (confocal and STED) of proteins labeled with nanobodies. A 775 nm depletion wavelength was used which is usually less practical for STED with fluorescent proteins.

Page 2L62 “there a few examples” à”there are few examples”

Page 2 L82 “depend on” à “depends on”

- Comparison between confocal and STED imaging demonstrates convincingly that nanobodies internalize in cells successfully and target efficiently DivIVA protein. Depending on the target protein, the cellular background is affected. Direct confocal imaging of fused proteins shows backgrounds of 40% (GFP) and 35% (RFP).  These are close to backgrounds observed in STED setup, with exception for NBGFP-Atto647n which is better in STED setup, this is difficult to rationalize (this binder seems to recognize specifically DivIVA involved in division septum).

- It is also very informative that the nature of the dye modifies the signal to background ratio with Atto594 being the least efficient (may be due to its lower emission wavelength and spectral overlaps with mCherry2).

- Authors should draw more attention about why it is beneficial to add a fluorescent label to fluorescent proteins. In particular, in the case of red fluorescent proteins; is it possible to get STED images without labelling with dyes at what wavelength ? what is the limitation of fluorescent proteins alone ? This should underline why it is decisive to use this method which implies cell membrane permeation (with the drawbacks of affecting biological process and of being more “work-demanding”). It could be interesting to get/(or try to get images) with fused proteins alone to have the comparison.
